METHODS AND RESOURCES

# AlphaFold2-multimer guided high-accuracy prediction of typical and atypical ATG8-binding motifs

**Tarhan Ibrahim**[1], **Virendrasinh Khandare**[1,2], **Federico Gabriel Mirkin**[1,3], **Yasin Tumtas**[1], **Doryen Bubeck**[1]\*, **Tolga O. Bozkurt**[1]\*

**1** Department of Life Sciences, Imperial College London, London, United Kingdom, **2** Department of Agrotechnology and Food Sciences, Biochemistry, Wageningen University and Research, Wageningen, the Netherlands, **3** INGEBI-CONICET, Ciudad Autonoma de Buenos Aires, Buenos Aires, Argentina

\* d.bubeck@imperial.ac.uk (DB); o.bozkurt@imperial.ac.uk (TOB)

**Data Availability Statement:** All relevant data are within the paper and its Supporting Information files. AF2-multimer predictions are uploaded to the

## Abstract

Macroautophagy/autophagy is an intracellular degradation process central to cellular homeostasis and defense against pathogens in eukaryotic cells. Regulation of autophagy relies on hierarchical binding of autophagy cargo receptors and adaptors to ATG8/LC3 protein family members. Interactions with ATG8/LC3 are typically facilitated by a conserved, short linear sequence, referred to as the ATG8/LC3 interacting motif/region (AIM/LIR), present in autophagy adaptors and receptors as well as pathogen virulence factors targeting host autophagy machinery. Since the canonical AIM/LIR sequence can be found in many proteins, identifying functional AIM/LIR motifs has proven challenging. Here, we show that protein modelling using Alphafold-Multimer (AF2-multimer) identifies both canonical and atypical AIM/LIR motifs with a high level of accuracy. AF2-multimer can be modified to detect additional functional AIM/LIR motifs by using protein sequences with mutations in primary AIM/LIR residues. By combining protein modelling data from AF2-multimer with phylogenetic analysis of protein sequences and protein–protein interaction assays, we demonstrate that AF2-multimer predicts the physiologically relevant AIM motif in the ATG8-interacting protein 2 (ATI-2) as well as the previously uncharacterized noncanonical AIM motif in ATG3 from potato (*Solanum tuberosum*). AF2-multimer also identified the AIM/LIR motifs in pathogen-encoded virulence factors that target ATG8 members in their plant and human hosts, revealing that cross-kingdom ATG8-LIR/AIM associations can also be predicted by AF2-multimer. We conclude that the AF2-guided discovery of autophagy adaptors/receptors will substantially accelerate our understanding of the molecular basis of autophagy in all biological kingdoms.

## Introduction

To withstand stressful conditions, eukaryotic cells employ a fundamental intracellular catabolic process known as macroautophagy or autophagy. Autophagy is central to cellular

public repository figshare and is available at https://doi.org/10.6084/m9.figshare.21533769.v1.

**Funding:** This work was funded by the Biotechnology and Biological Sciences Research Council (BBSRC, UK) BB/T006102/1. T.O.B and Y. T. was funded by the BBSRC grant (BB/T006102/1). T.I. was funded by a BBSRC-DTP PhD studentship (BB/M011178/1), F.G.M. was funded by a British Society for Plant Pathology Incoming Fellowships. The funders had no role in study design, data collection and analysis, decision to publish, or preparation of the manuscript.

**Competing interests:** I have read the journal's policy and the authors of this manuscript have the following competing interests: T.B receives funding from industry on NLR biology. TB is a founder and shareholder at Resurrect Bio Ltd.

**Abbreviations:** AF2, AlphaFold2; AIM/LIR, ATG8/LC3 interacting motif/region; hRTN3, human RTN3; IDPR, intrinsically disordered protein region; PCR, polymerase chain reaction; pLDDT, predicted local distance difference test; RMSD, root mean square deviation; SDM, site-directed mutagenesis; SLIM, short linear motif.

homeostasis in all eukaryotes, from humans to yeasts and plants since it mediates adaptation to harmful environmental conditions by eliminating damaged and toxic cellular components as well as invading microbes. Autophagy plays critical roles in various physiological and pathological conditions, particularly defence against pathogens, thus directly affecting plant and human health [1].

Autophagy is a multistep process initiated by the induction of an isolation membrane that expands and closes to form a double-membrane vesicle named the autophagosome. Autophagy cargoes are typically loaded into the inner leaflet of the isolation membrane [2]. Mature autophagosomes are transported to fuse with lysosomes, which in turn digest the captured cargoes [3]. Alternatively, some autophagosomes are re-routed towards the cell surface to discharge cytoplasmic components outside the cell, through a process called secretory autophagy [4]. In yeast (*Saccharomyces cerevisiae*), autophagy is coordinated by more than 30 core proteins known as the ATG (autophagy-related) proteins that are recruited to autophagosome formation sites in a hierarchical manner [2]. Many of these proteins are also conserved in plants and humans, such as the ubiquitin-like ATG8-family proteins that play central roles in virtually all steps of autophagy, from cargo sequestration to transport and lysosomal fusion of autophagosomes [5,6]. Although yeast has only a single copy of ATG8, the protein has diversified into multiple isoforms in plants (ATG8A–I) and humans (referred to as the LC3/GABARAP family), forming family-specific ATG8 clades [7]. ATG8 associates with proteins that regulate autophagy initiation, such as ATG7, ATG3, and ATG1 [8,9], and decorates both inner and outer autophagosomal membranes to coordinate autophagosome biogenesis and transport. ATG8 sequesters specific cargoes inside the autophagosomes by interacting with autophagy cargo receptors that capture specific cargoes [10,11]. In addition, ATG8 binds to autophagy adaptors on the autophagosome surface to modulate autophagosome transport and fusion events [12,13].

ATG8-interacting proteins carry a short linear motif (SLIM) called the ATG8/LC3 interaction motif/region (AIM/LIR) that interacts with the core autophagy machinery as well as autophagy adaptors and receptors [14]. The core AIM/LIR motif sequence ([W,Y,F][X][X][L,I,V]) consists of an aromatic amino acid followed by any 2 amino acids and a hydrophobic residue. The first aromatic and last hydrophobic amino acids of the core AIM/LIR motif bind to 2 hydrophobic pockets—also known as W (hydrophobic pocket 1) and L (hydrophobic pocket 2) pockets—on the surface of an ubiquitin-like fold in ATG8 proteins. The sequences flanking the aromatic residue of the AIM/LIR typically consist of negatively charged residues, which enhance AIM/LIR docking by forming polar interactions with the positively charged residues surrounding the W and L pockets [15]. ATG8 binding can be improved further via posttranslational modifications, such as phosphorylation of the residues flanking the core AIM/LIR regions [16,17]. However, noncanonical AIM/LIR motifs that mediate ATG8 binding have also been discovered, expanding the spectrum of residues that facilitate ATG8 association [18,19]. The emerging paradigm is that autophagy is primarily orchestrated through sequential binding of core autophagy components, autophagy adaptors, and cargo receptors to the ATG8-family members via canonical/noncanonical AIM/LIR motifs. Therefore, identifying and characterising the AIM/LIR residues has been a crucial step in dissecting the molecular basis of autophagy regulation from initiation to selective cargo sorting and autophagosome transport, which may also be important for understanding newly discovered ATG8ylation process [20,21].

As the canonical AIM/LIR sequence consensus occurs in many proteins, identifying functional AIM/LIR motifs has proven challenging. Thus, a simple search for the [W,Y,F][X][X][L,I,V] amino acid sequence predicts many false AIM/LIR motifs. Since proteins of interest might also have multiple potential AIM/LIR motifs that match the consensus pattern, these

motifs must be validated through mutagenesis, peptide binding, and autophagy assays. For instance, the yeast ATG7 has 16 residues that match the core AIM/LIR consensus sequence, yet none have been functionally validated. Likewise, the human NDP52 autophagy cargo receptor has 8 predicted AIM/LIRs, but binds to LC3 through a different, noncanonical LIR [18]. However, a distinctive feature of the AIM/LIR motif that can help narrow down the number of candidates has been discovered: functionally validated AIM/LIR motifs are typically located in intrinsically disordered protein regions (IDPRs) [14]. To help discover functional AIM/LIRs, in silico AIM/LIR prediction tools such as iLIR [22], hfAIM [23], and pLIRm [24] can also be used. Although these computational tools can be used somewhat successfully to predict canonical AIM/LIR motifs, none can detect noncanonical AIM/LIR motifs. Additionally, none of these methods determine the spatial distribution of AIM/LIR motifs or flanking residues on ATG8 proteins; however, this level of resolution combined with the ability to determine side chain interactions established by the flanking AIM/LIR residues would significantly improve investigations into ATG8 binding and help determine the ATG8-binding specificity of various autophagy-related proteins. Furthermore, these tools would accelerate studies aiming to dissect the evolutionary dynamics of autophagy regulation in eukaryotes.

The single-chain protein structure prediction tool AlphaFold2 (AF2) [25], and the recently retrained AlphaFold-Multimer (AF2-multimer) system that can predict homomeric and heteromeric interfaces [26], has sparked a renewed interest in applying structural biology to understanding complex cellular processes. In this study, we investigated whether AF2-multimer could identify ATG8-binding structures mediated by AIM/LIR motifs carried by various proteins that have been shown to bind ATG8. AF2-multimer showed 90% accuracy in determining AIM/LIR motifs in 33 experimentally validated proteins that carry functional AIM/LIR motifs. At present, AF2-multimer can identify multiple AIM/LIR residues even after in silico deletion/mutagenesis of the primary AIM/LIR motif. Strikingly, AF2-multimer predicted all 3 noncanonical AIM/LIR motifs that are experimentally validated, as well as the previously uncharacterized AIM motif in plant ATG3, which is not possible with other current prediction tools. Furthermore, AF2-multimer predicted functional AIM/LIRs in 3 of the 4 tested proteins encoded by plant and human pathogens that target host ATG8 proteins, indicating that AF2-multimer can also detect cross-kingdom AIM/LIR–ATG8 interactions. Our study highlights the potential of AF2-multimer in identifying AIM/LIR residues and provides a framework for discovery of new autophagy receptors and adaptors. This AI-guided approach will substantially accelerate our understanding of the molecular basis of autophagy in all kingdoms of life.

## Results

### AF2-multimer predicts experimentally validated canonical AIM/LIR motifs with high accuracy

To determine the extent to which AF2-multimer predicts AIM/LIR motifs, we decided to run AF2-multimer using a subset of proteins that carry experimentally validated, canonical (W/Y/F-X-X-L/I/V) AIM/LIR motifs along with their corresponding ATG8 proteins from plants, humans, and yeast. The AF2-predicted models of plant ATG8CL, human GABARAP, and yeast ATG8 proteins display highly similar backbone and side chain conformations to previously solved crystal structures [27–29]. All tested ATG8 AF2-models have root mean square deviation (RMSD) scores close to 0.5Å (*Sc*ATG8: 0.51Å; *Hs*GABARAP: 0.51Å; *St*ATG8CL: 0.50Å), suggesting AF2-multimer can be used to identify ATG8-binding proteins. Reflecting the conserved nature of the AIM/LIR consensus sequence, amino acid alignments between yeast, plant, and human ATG8 isoforms showed that amino acids forming the W- (site 1) and

L- (site 2) pockets, which accommodate the AIM/LIR motifs, show high sequence similarity (Fig 1A and 1B). This observation suggests that universal binding patterns govern ATG8–LIR/AIM associations and that these patterns can be predicted by AF2-multimer. Among the 36 tested proteins with experimentally validated canonical ([W,Y,F][X][X][L,I,V]) AIM/LIR motifs, AF2-multimer predicted 33 (approximately 90%) of them accurately (Figs 1C–1H and S1A–S1F). Consistent with previous work that revealed canonical AIM/LIRs are typically found in structurally disordered regions [14], nearly all AF2-multimer predicted AIM/LIRs appeared in IDPRs, while a few were predicted to be partially disordered. These findings are consistent with the view that AIM/LIR motifs are structurally flexible protein interaction interfaces, referred to as SLiMs, that display dynamic structural plasticity to precisely associate with their substrates [14].

Despite its high accuracy, AF2-multimer failed to detect AIM/LIR motifs in plant Joka2 (also known as NBR1) and human Calreticulin proteins [30,31]. The confidence scores determined by the predicted local distance difference test (pLDDT) [25] in Joka2's AIM motif were low (approximately 50) (S1H Fig), which could account for the failure of AIM/LIR prediction. Although confidence scores in the previously identified LIR motif of Calreticulin were high (>70), AF2-multimer did not identify any interaction between LIR and LIR docking (LDS) regions (S1G Fig). Superimposition of the Calreticulin crystal structure (3POW) [32] with the AF2-multimer model showed a high correlation (RMSD: 0.33Å), indicating that AF2-multimer's inability to identify the functional LIR is not a consequence of structural mis-prediction. Of note, the AIM/LIR motifs in Joka2 and Calreticulin are embedded in short, disordered regions (5 and 6 amino acids long, respectively) that are flanked by alpha-helices and beta-strands, respectively. The flexibility of the AIM/LIR motifs may be constrained by these ordered regions, which could hinder ATG8 docking. We speculate that AF2-multimer predictions might fail in such cases because the AIM/LIR–ATG8 association is somewhat conditional, since posttranslational modifications, intramolecular interactions, or interactions with other proteins can influence ATG8 binding [16,33,34].

Intriguingly, AF2-multimer revealed a distinct canonical LIR motif (FVDV) in the human autophagy substrate DVL2 that differed from the one previously discovered (WLKI) (S1I Fig). Mutating DVL2$^{WLKI}$ to DVL2$^{ALKA}$ reduced, but did not prevent, LC3 binding [35], indicating that additional LIR residues, as predicted by AF2-multimer, may influence LC3 binding. The new LIR predicted by AF2-multimer (FVDV) is in the C-terminal IDPR, as opposed to the previously verified LIR motif (WLKI), which is situated between 2 beta-strands (S1J Fig). This may account for AF2-multimer's preference for the second LIR. Nevertheless, further research is needed to determine whether the additional LIR predicted by AF2-multimer is valid. Thus, AF2-multimer is a powerful tool for predicting canonical AIM/LIR motifs with a high level of accuracy; however, several limitations, such as conditional ATG8 binding due to posttranslational modifications, may limit its effectiveness.

## AF2-multimer can predict multiple AIM/LIR motifs

The human E3 ubiquitin ligase NEDD4 was previously reported to carry 2 LIR motifs that bind LC3B. Deletion analysis of NEDD4 revealed that LIR1 (WVVL) is not necessary for LC3 binding, whereas deletion of both LIR1 and LIR2 (WEII) impairs LC3 binding [36]. Although LIR2 is more critical for LC3 binding, it is possible that LIR1 is also functional. Interestingly, AF2-multimer predicted LIR1 over LIR2 for AIM pocket association (Fig 2A). To determine whether AF2-multimer could also predict LIR2, we truncated LIR1 and re-ran AF2-multimer prediction [36]. In line with previous findings, LIR2 displayed a perfect association with the AIM pocket in the absence of LIR1 (Fig 2B). By contrast, an in silico truncated NEDD4

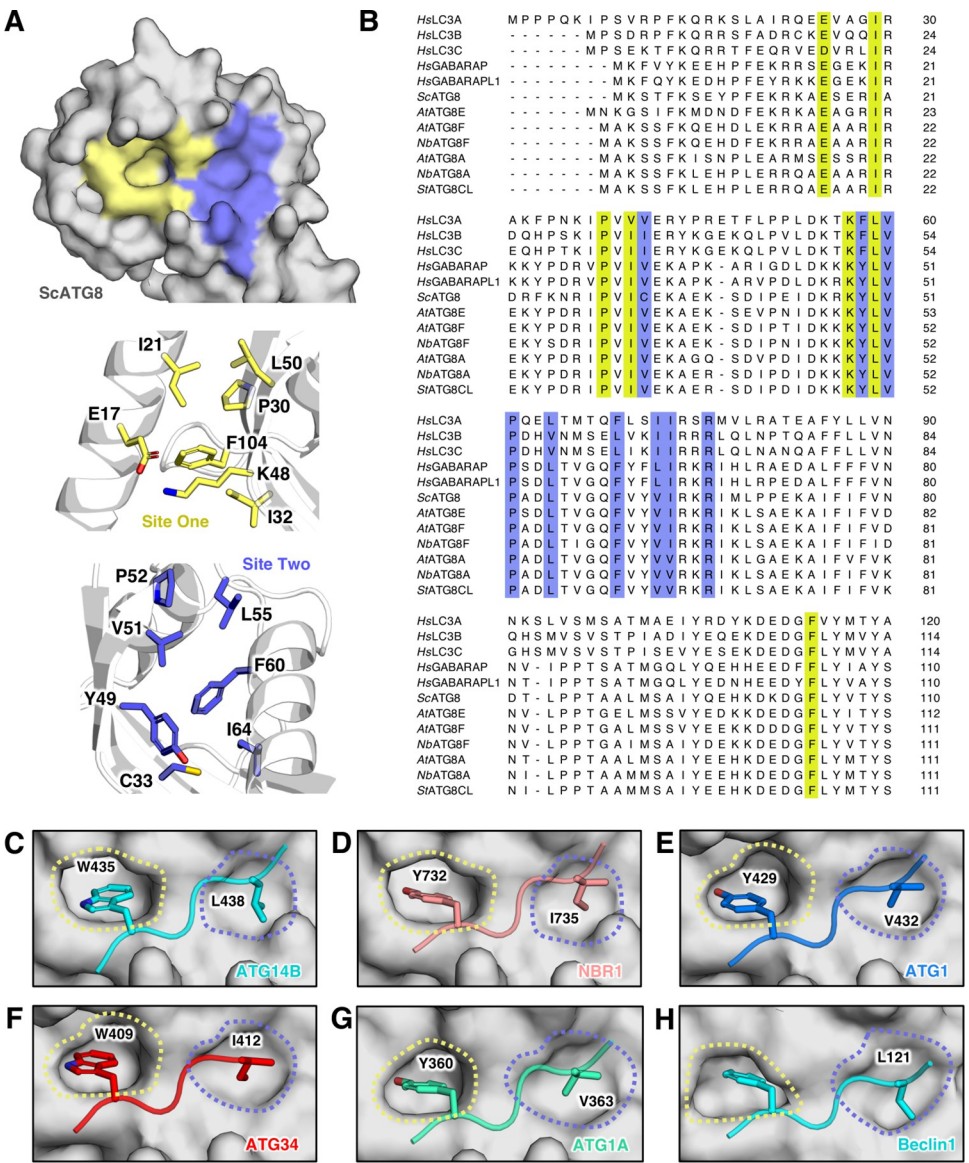

**Fig 1. AF2-multimer can correctly identify experimentally validated canonical AIM/LIRs in ATG8-interacting proteins. (A)** The yeast ATG8 (surface representation–grey) has 2 AIM/LIR pockets, known as W-site (site 1 –yellow) and L-site (site 2 –slate blue) (top panel). The core of the pockets is formed by residues phenylalanine (F), leucine (L), valine (V), isoleucine (I), tyrosine (Y), proline (P) and polar amino acids glutamate (E), lysine (K), and cysteine (C) guard the pocket. Residues comprising site 1 and site 2 are shown in the middle and bottom panels, respectively. **(B)** These residues are conserved among the ATG8-family of all species used for this study. Residues conserved and forming the W- (yellow) and L- (slate blue) sites have been highlighted accordingly. Some examples of accurate AF2 multimer validation of experimental work include human GABARAP interactions with **(C)** ATG14B (cyan) and **(D)** NBR1 (salmon), yeast ATG8 interactions with **(E)** ATG1 (blue) and **(F)** ATG34 (red), **(G)** *A. thaliana* ATG8A with ATG1A (green) and **(H)** *N. benthamiana* ATG8A with Beclin1 (cyan). Residues interacting with the AIM/LIR pockets identified by AF2 multimer and experimentation have been highlighted via their 1 letter amino acid code and position within the protein and shown with their side chains. Side chains have been coloured based on elements by PyMOL, where oxygen is red and nitrogen is blue. Data are available in S1 Data. Models coloured based on AF2-multimer prediction confidence are available in S1 Fig. AF2-multimer prediction files are available from https://doi.org/10.6084/m9.figshare.21533769.v1. AF2, AlphaFold2; AIM/LIR, ATG8/LC3 interacting motif/region.

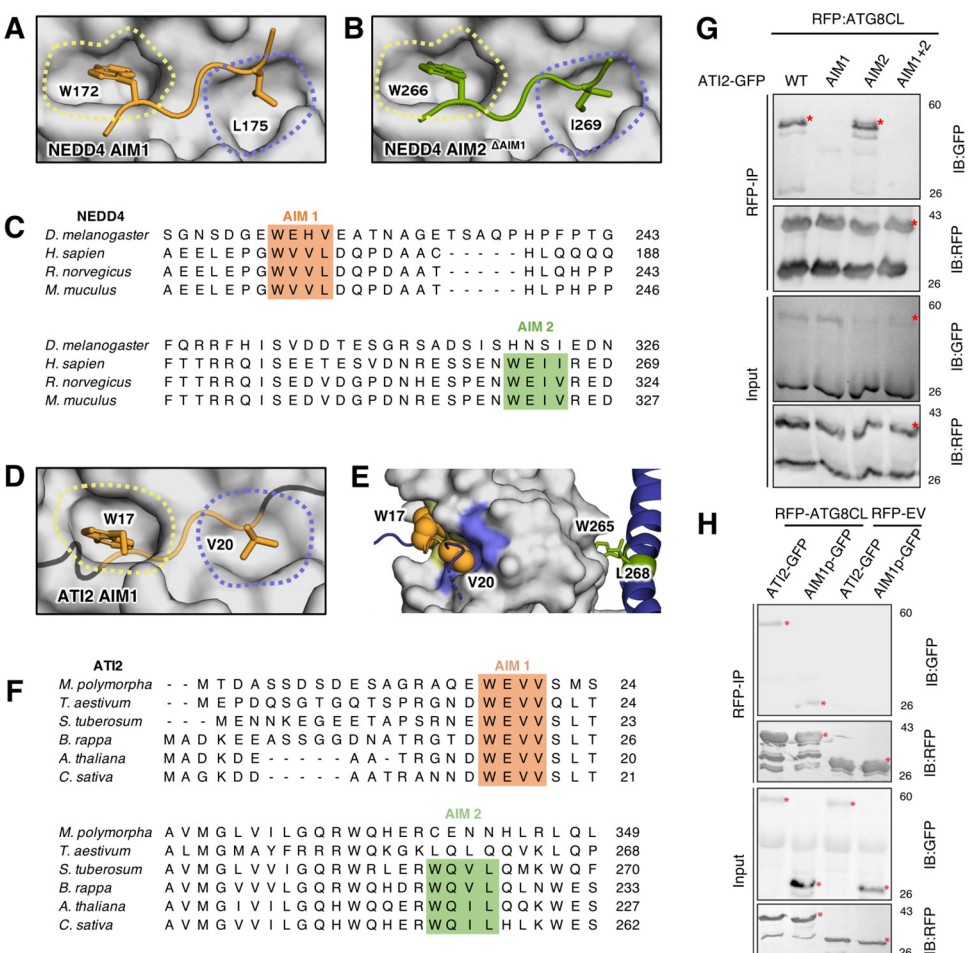

**Fig 2. AF2 multimer can correctly identify a functional AIM/LIR among various targets. (A)** AF2-multimer predicts AIM1 (172)WxxL(175) of NEDD4 as the interactor for LC3B pockets. **(B)** Upon the truncation of AIM1, AF2-multimer predicts another interface at (266)WxxI(269), which was identified as AIM2, previously. **(C)** Two AIMs of NEDD4 are conserved among *Drosophila melanogaster*, *Homo sapiens*, *Rattus norvegicus*, and *Mus musculus*. **(D)** ATI2 contains two-functional AIMs and AF2-multimer predicts the interface at AIM1 (17)WxxV(20) as the functional AIM. **(E)** AIM2 of ATI2, (265)WxxV(268), is localised in an alpha-helix and a distance away from the LC3B AIM pockets. **(F)** AIM1 of ATI2 is conserved among *Marchantia polymorpha*, *Triticum aestivum*, *Solanum tuberosum*, *Brassica rapa*, *Arabidopsis thaliana*, and *Cannabis sativa*, whereas AIM2 is not conserved in *M. polymorpha* and *T. aestivum*. AIM1 and AIM2 for both NEDD4 and ATI2 have been coloured as orange and green, respectively. (G) In planta co-immunoprecipitation between ATG8CL and ATI2 and its mutants (ATI2AIM1, ATI2AIM2, or ATI2AIM1 +2). RFP-ATG8CL was transiently co-expressed with either ATI2-GFP, ATI2$^{AIM1}$GFP, ATI2$^{AIM2}$-GFP, or ATI2$^{AIM1}$$^{+2}$-GFP. Red asterisks indicate expected band sizes. (H) In planta co-immunoprecipitation between ATG8CL and ATI2 and ATI2 AIM1 peptide. RFP-ATG8CL or RFP-EV control was transiently co-expressed with either ATI2-GFP or ATI2 AIM1p-GFP. Red asterisks indicate expected band sizes. Data are available in S1 Data. Models coloured based on AF2-multimer prediction confidence are available in S2 Fig. AF2-multimer prediction files are available from https://doi.org/10.6084/m9.figshare.21533769.v1. Raw images of this figure are provided in S1 Raw images. AF2, AlphaFold2; AIM/LIR, ATG8/LC3 interacting motif/region.

sequence lacking both LIRs failed to form any association with the AIM pocket on LC3B (S2A Fig). Consistent with our AF2-predictions, a different study showed that both LIR1 and LIR2 peptides bind LC3B with similar KDs [37]. Phylogenetic analysis revealed that both LIRs are conserved, further supporting the AF2-multimer predictions that NEDD4 has 2 functional LIR motifs (Fig 2C). To determine whether AF2-multimer predicts LIR1 over LIR2 simply because LIR1 is positioned upstream of LIR2, we swapped the position of LIR2 ($X_4WEEIX_4$)

with that of LIR1 ($X_4$WVVL$X_4$) and re-ran AF2-multimer. After this change was made, AF2-multimer predicted that LIR2 was the main functional LIR motif (S2B Fig), indicating that AF2 prioritises the first functional AIM/LIR.

We next used this approach to assess the plant ATG8-interacting protein-2 (ATI-2), which has 2 predicted AIM motifs [38]. A recent yeast-two-hybrid analysis of ATI-2 and ATG8f proteins from the model plant *Arabidopsis* (*Arabidopsis thaliana*) indicated that the N-terminal AIM1 in ATI-2 is necessary and sufficient for ATG8 interaction [39]. We examined whether AF2-multimer could accurately predict the functional AIM in the potato (*Solanum tuberosum*) ATI-2 protein. Consistent with the experimental evidence provided for the *Arabidopsis* ATI-2 homologue [39], AF2-multimer identified AIM1 of the potato ATI-2 as associating with the AIM pocket on the ATG8CL variant (Fig 2D and 2E). In silico truncation of the core AIM1 residues ablated any AIM2–ATG8 association as predicted by AF2-multimer (S2F Fig). Multiple sequence alignments revealed that although the potato AIM1 sequence is conserved, the AIM2 sequence is only conserved in closely related dicot plants but not in distantly related plant species, suggesting that AIM2 may not be a functional AIM (Fig 2F). To validate this AIM prediction, we tested the interaction of potato ATG8CL with ATI-2 constructs harbouring AIM1, AIM2, and AIM1,2 double mutations in plant cells. In agreement with the AF2-multimer predictions, our co-immunoprecipitation experiments showed that ATI-2AIM1 and ATI-2AIM1,2 double mutants fail to interact with ATG8CL, whereas the ATI-2AIM2 mutant binds ATG8CL much like the wild-type ATI-2 protein (Fig 2G). We further validated these findings by generating an AIM1 peptide construct that effectively bound ATG8CL in planta (Fig 2H). These results indicate that while AF2-multimer prioritises the first functional AIM/LIR in a protein containing several functional AIM/LIR motifs, it prevents the prediction of spurious AIM/LIRs. Thus, an in silico protein truncation approach can be used to identify multiple AIM/LIRs using AF2-multimer. To test the ability of AF2-multimer in the identification of AIM/LIR proteins with more than 2 candidates, we investigated human RTN3 (hRTN3) that reports 6 LIR motifs [40]. Running AF2-multimer with hRTN3 showed a canonical LIR at the third position (FEVI), in agreement with structural data [41] (S2G Fig). In silico, truncation of this LIR did not show any additional LIR interactions with its partner GABARAP-L1. Interestingly, only this LIR is conserved, which may indicate a limitation of our phylogenetic-based AI approach.

## AF2-multimer can predict noncanonical AIM/LIRs

Since there are no established techniques to predict noncanonical AIM/LIR motifs, their discovery poses a challenge [42]. We investigated the ability of AF2-multimer to recognise previously characterised noncanonical LIR motifs in the human autophagy cargo receptors NDP52 and TAX1BP1 [18,43]. AF2-multimer accurately identified the validated noncanonical LIR motifs of NDP52 (ILVV) and TAX1BP1 (MLVV) with high confidence. In these 2 resolved models, I/M residues occupy the W pocket (site 1), while the final Vs are located in the L pocket (site 2) of their respective LC3 proteins. Central L and V residues, however, appear to form additional hydrophobic surface interactions (Fig 3A and 3B). We also used AF2-multimer to predict the yeast ER-phagy cargo receptor ATG40, which carries a noncanonical AIM (YDFM) with an oddly positioned M in the last AIM position rather than the standard L/I/V residues [41]. AF2-multimer accurately predicted the noncanonical AIM in this case as well and showed that the atypical M residue occupies the L pocket on ATG8 as expected.

Another protein that was reported to have a noncanonical AIM/LIR is the *Arabidopsis* C53 (AtC53), which binds to ATG8 through several shuffled AIMs with the consensus sequence IDWG/D in addition to the canonical AIM YEIV [44,45]. AF2-multimer analysis using AtC53

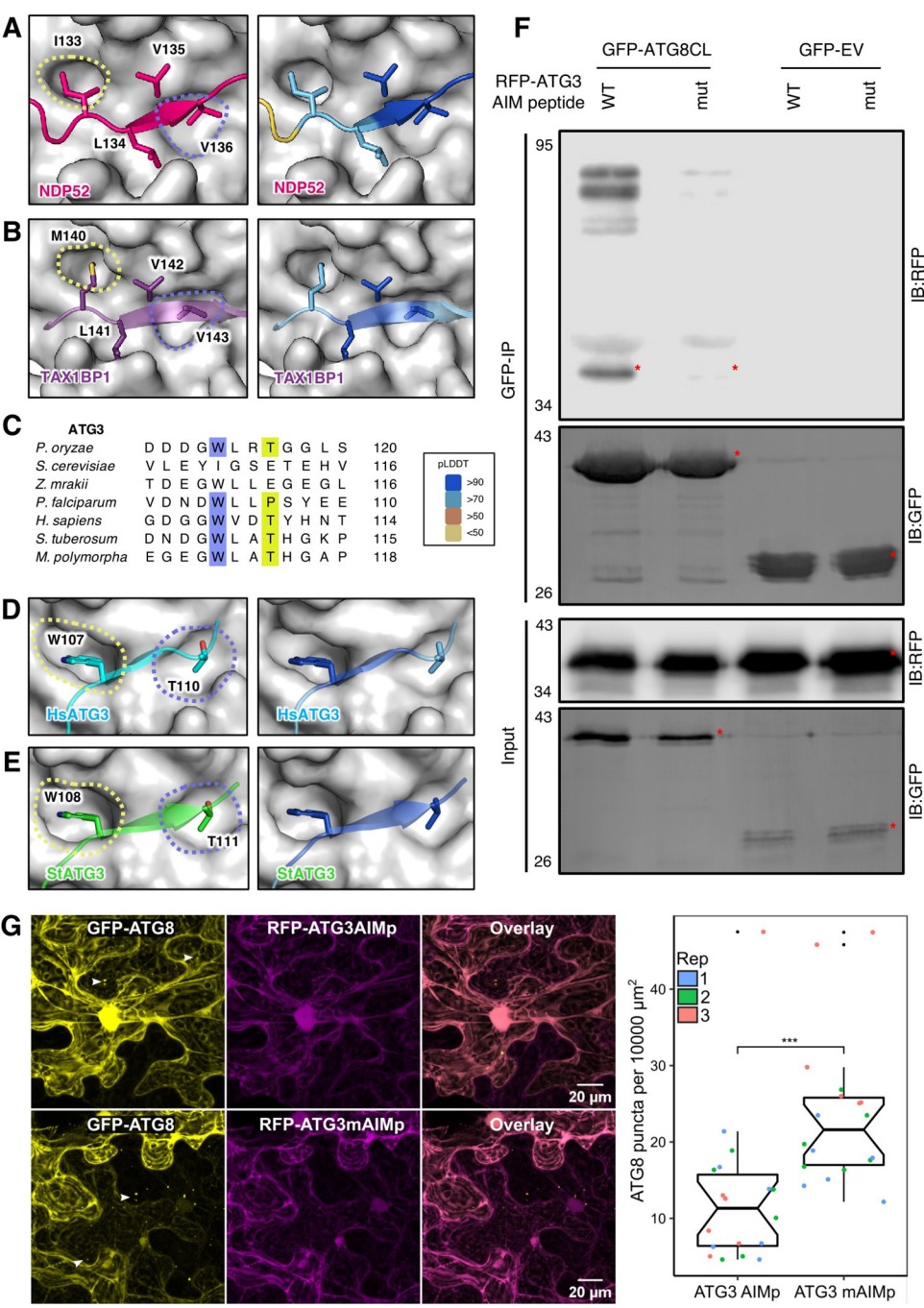

**Fig 3. AF2-multimer can accurately predict experimentally validated noncanonical AIM/LIRs. (A)** NDP52 LIR in complex with LC3C. Beyond the LIR pockets, W- (yellow) and L- (slate) sites, NDP52 (dark pink) utilises resides LEU134 and VAL135 to form additional interactions to provide specificity, as shown in literature. AF2-multimer predicts this complex with high confidence. **(B)** TAX1BP1 LIR in complex with LC3A. Beyond the LIR pockets W- (yellow) and L- (slate) sites, TAX1BP1 (purple) utilises resides LEU141 and VAL142 to form additional interactions to provide specificity, as shown in literature. AF2-multimer predicts this complex at a high confidence. **(C)** The noncanonical LIR of ATG3 is conserved in all but 2 species studied for this project. **(D, E)** ATG3 (human–cyan, *S. tuberosum*–green) AIM/LIR in complex with ATG8-family members. Residues TRP and THR are arranged in a noncanonical AIM/LIR, as shown in literature. Right-hand side of panels A, B, D, and E are coloured based on the AF2-calculated prediction confidence score; pLDDT. Blue indicates regions of a protein with a score of over 90, meaning a very high confidence prediction. Scores between 70 and 90 are represented with light blue, which accounts for a high confidence score. Scores between 50 and 70 are considered low (orange) and anything below 50 is a very low confidence prediction (yellow). **(F)** In planta co-immunoprecipitation between ATG8CL and ATG3 AIM peptide and

its mutant (WT/mut). GFP-ATG8CL or empty vector GFP control was transiently co-expressed with either RFP-wtAIMp or RFP-mutAIMp. Red asterisks indicate expected band sizes. **(G)** Confocal microscopy of wild-type *N. benthamiana* co-expressing GFP-ATG8CL and RFP-ATG3AIMpeptide (WT or mutant). White arrows indicate example puncta that was counted with different sizes. Images shown are maximal projections of 18 frames with 1 μm steps. Quantification of ATG8CL autophagosome puncta per 10,000 μm$^2$ with WT and mutant (M) ATG3 AIM revealed a significantly higher number of puncta in sample co-expressed with mutated peptide compared to WT. Each repeat for the microscopy and quantification is represented by a different colour. Data are available in S1 Data. Models coloured based on AF2-multimer prediction confidence are available in S3 Fig. AF2-multimer prediction files are available from https://doi.org/10.6084/m9.figshare.21533769.v1. Raw images of this figure are provided in S1 Raw images. Individual numerical values and code for statistical analysis for panel G are available in S2 Data and S3 Data, respectively. AF2, AlphaFold2; AIM/LIR, ATG8/LC3 interacting motif/region; pLDDT, predicted local distance difference test.

revealed the canonical AIM YEIV (S3A and S3B Fig). To determine whether AF2-multimer can predict the noncanonical AIM in AtC53, we re-ran the analysis using an AtC53 sequence in which the canonical AIM had been mutated from YEIV to AEIA. In the absence of this canonical AIM, AF2-multimer was able to predict 1 of the 3 shuffled AIMs in C53 with the sequence WDVSV (amino acids 335–339) [44,45]. According to the AF2-multimer model, while the W occupies the expected W-site (site 1) on AtATG8a, the second V at position 339 occupies the L-site (site 2), suggesting the existence of an extended noncanonical AIM with the sequence WDVSV (S3C and S3D Fig). To test the limitations and capabilities of AF2-multimer, we next explored whether the noncanonical LIR in ATG12 could plausibly bind ATG8 [46]. Although AF2-multimer did not predict any AIM-dependent interaction between these 2 proteins, a possible additional interaction interface was highlighted, which requires further investigation.

Encouraged by the accurate prediction of canonical/noncanonical AIM/LIR motifs, we next checked if AF2-multimer could identify previously uncharacterized AIM//LIR residues. We decided to test ATG3 because, although it is well established that it binds ATG8, how it does so remains controversial in different organisms [47]. While a canonical AIM is reported to mediate ATG8 binding in yeast, this AIM is conserved in fungi but not in human and plant ATG8s (Fig 3C). Intriguingly, AF2-multimer predicted a noncanonical AIM/LIR in both plant and human ATG3 proteins with WXXT sequence (Fig 3D and 3E). Consistent with the AF2 predictions, ATG3 from the malaria parasite (*Plasmodium falciparum*) was found to have a noncanonical AIM in this region with the WLLP sequence [48]. To validate the AF2-multimer prediction of the ATG3-ATG8 interaction, we cloned the AIM peptide region ($X_{17}WLATX_4$) of the potato ATG3 (StATG3) and its AIM mutant ($AX_{17}ALAAX_4$) and performed pull-down assays with the potato ATG8CL (StATG8CL) in plant cells. This revealed that wild-type ATG3 AIM peptide binds ATG8CL, whereas the mutated AIM peptide shows a significant reduction in ATG8CL interaction, validating the AF2-predicted model of noncanonical AIM in plant ATG3 (Fig 3F). We have previously demonstrated that overproduction of functional AIM peptides blocks autophagosome formation in plants [49], which could be used as proxy to estimate AIM peptide ATG8 interaction in planta. Overexpression of the ATG3 AIM peptide reduced the formation of ATG8CL puncta (Fig 3G), indicating that it can block autophagy when overproduced in plant cells. In agreement with our findings, a recent report showed human ATG3 interacts with LC3 through the noncanonical LIR sequence WXXT [50]. These results show that the AF2-multimer can predict both canonical/noncanonical AIM/LIRs and can serve as a powerful tool when combined with the molecular phylogeny analysis to provide new autophagy understanding.

## AF2-multimer predicts cross-kingdom AIM/LIR-ATG8 interactions in host–pathogen interactions

Since many plants and human pathogens deploy virulence factors (also known as effectors) that target the host autophagy machinery, we investigated whether AF2-multimer can identify

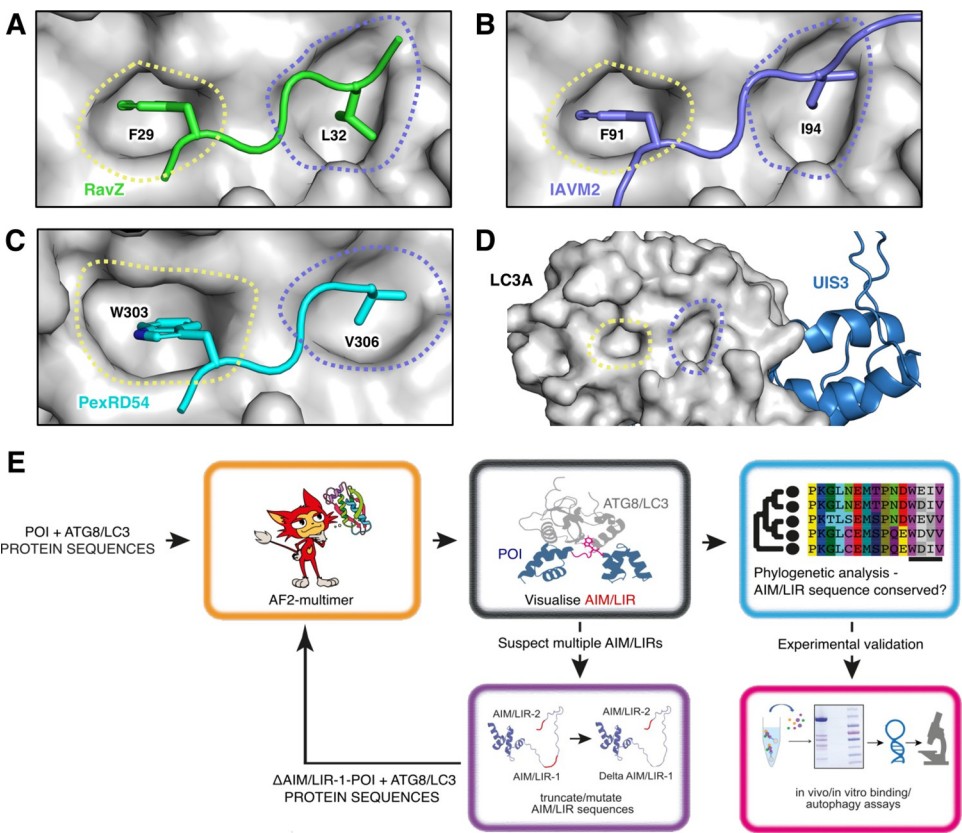

**Fig 4. AF2-multimer guided discovery of cross-kingdom interactions. (A–C)** RavZ (Legionella—green), M2 (Influenza–slate blue), and PexRD54 (*P. infestans*–cyan) interact with their respective ATG8-family member in humans (RavZ, M2) and plants (PexRD54) to produce a pathogenic phenotype. AF2-multimer correctly predicts interactions studied in literature. **(D)** UIS3 (*Plasmodium*–dark blue) does not interact with LC3A via its AIM/LIR pockets. **(E)** The pipeline for AF2-multimer and phylogeny guided discovery and validation of AIM/LIRs among various species. POI: protein of interest. Data are available in S1 Data. Models coloured based on AF2-multimer prediction confidence are available in S3 Fig. AF2-multimer prediction files are available from https://doi.org/10.6084/m9.figshare.21533769.v1. AF2-multimer icon by Mirdita and colleagues [55], used under CC BY 4.0 licence (https://s100.copyright.com/AppDispatchServlet?title=ColabFold%3A%20making%20protein%20folding%20accessible%20to%20all&author=Milot%20Mirdita%20et%20al&contentID=10.1038%2Fs41592-022-01488-1©right=The%20Author%28s%29&publication=1548-7091&publicationDate=2022-05-30&publisherName=SpringerNature&orderBeanReset=true&oa=CC%20BY). AF2, AlphaFold2; AIM/LIR, ATG8/LC3 interacting motif/region.

cross-kingdom ATG8–AIM/LIR associations in host–microbe interactions. To do so, we ran AF2-multimer using 3 pathogen virulence factors that bind host ATG8/LC3 proteins via experimentally validated AIM/LIR motifs. This included the RavZ protein secreted into human cells by the bacterial pathogen *Legionella pneumophila* [51], Matrix protein 2 (M2) deployed by the Influenza A virus [52], and the PexRD54 protein secreted into plant cells by the Irish potato famine pathogen *Phytophthora infestans* [53]. For all 3 pathogen proteins carrying canonical AIM/LIRs, AF2-multimer successfully predicted the correct residues that mediate ATG8 interactions (Figs 4A–4C and S3). AF2-multimer accurately predicted the functional C-terminal AIM (WEIV) in PexRD54 from among the 2 potential AIMs and the functional N-terminal LIR (FDLL) in RavZ from among 8 candidate LIRs, once again demonstrating the ability of AF2-multimer to determine the correct AIM from among multiple candidates. Of note, in the case of M2, the highest ranked AF2-model does not show a proper LIR (FVSI) fit in the LDS, whereas other models do show a perfect alignment of the

core LIR residues, F and I, on the corresponding hydrophobic pockets on LC3. This is most likely because the confidence score of the first ranked model in the LIR/LDS region was much lower than that of other models (S3 Fig), indicating that confidence scores around the AIM/ LIR region are more reliable than the overall protein model scores when determining AIM/ LIR motifs by AF2-multimer. Additionally, we analysed the malaria (*Plasmodium berghei*) virulence factor PbUIS3, which does not carry a canonical LIR motif but instead binds to LC3 through an unidentified cryptic interface [54]. However, none of the AF2-multimer models showed any association of PbUIS3 residues with LC3-LDS, instead suggesting a different binding interface. According to these models, the previously identified PbUIS3 residues (D173, Y174, D175, E205, and K209) that are reported to be important for LC3 binding [54] are embedded in 2 separate alpha-helices, which do not associate with LC3A (S3E Fig), suggesting that mutations in these loci might indirectly limit LC3 binding. It would be interesting to investigate whether the AF2-multimer prediction of PbUIS3-LC3 association is correct or not. Despite limitations such as conditional ATG8 binding that can prevent AIM/LIR predictions, these analyses show that AF2-multimer can predict cross-kingdom AIM/LIR–ATG8 interactions.

## Discussion

We provide evidence that AF2-multimer can predict both canonical and noncanonical AIM/ LIR motifs with high accuracy. In 36 proteins that carry at least 1 experimentally validated canonical AIM/LIR motif, AF2-multimer accurately identified 33 functional AIM/LIRs among 310 candidate sequences that match the canonical AIM/LIR consensus (Fig 1). AF2-multimer can also predict more than 1 functional AIM/LIR in a protein; however, it seems to prioritise displaying the N-terminal functional AIM/LIR. Nevertheless, this limitation can be circumvented by re-running AF2-multimer using truncated/mutated sequences of the first AIM/LIR (Fig 2). AF2-multimer accurately predicted 4 out of 5 experimentally validated noncanonical AIM/LIRs (Fig 3), an ability that is lacking in current AIM/LIR prediction algorithms. Remarkably, AF2-multimer accurately precited a previously uncharacterised noncanonical, functional AIM motif in plant ATG3, which we validated by ATG8 binding and autophagy assays (Fig 3). Furthermore, AF2-multimer performed well in identifying AIM/LIR motifs in 3 out of 4 tested pathogen virulence factors that target ATG8 members in their plant and human hosts, revealing that cross kingdom ATG8–LIR/AIM associations can also be predicted by AF2-multimer. AF2-multimer is therefore quite effective in identifying true AIM/LIR motifs among multiple other non-functional motifs in a protein of interest with high confidence, which minimises the experimental effort needed to validate these motifs. The confidence of these predictions can be strengthened by combining evolutionary information from phylogenetic analysis of protein sequences to determine AIM/LIR conservation, which should be considered when designing time-consuming validation experiments. The AF2-guided AIM/LIR prediction workflow consists of structural prediction of ATG8 and candidate proteins, phylogenetic analysis of predicted AIM/LIR motifs, and experimental validation by in vitro/in vivo binding assays (Fig 4E). If present, additional AIM/LIR residues can be identified by re-running the protocol using protein sequences that are mutated (in silico) in the primary AIM/LIR determined after the first run (Fig 4E).

A major advantage of AF2-multimer is that it provides spatial resolution of the AIM/LIR-- ATG8 interaction, displaying not only structural alignment of the LIR/AIM residues on LDS/ AIM docking site (ADS) but also additional associations through LIR/AIM flanking residues. This is important because it helps determine the extent to which AIM flanking residues condition AIM/LIR specificity towards certain ATG8 isoforms, thereby providing unprecedented

insights into evolutionary studies focusing on autophagy regulation. This is especially important for discovering the mode of action of noncanonical AIM/LIR motifs, which are often overlooked as there are no known distinctive features to predict them by. Remarkably, using AF2-mitimer combined with experimental approaches and phylogenetic analysis, we were able to identify and validate a previously uncharacterised functional, noncanonical AIM motif in an IDPR of the plant ATG3 (Fig 3), indicating the AF2-multimer can inform new biology. As reported before [14], an emerging unifying pattern of the canonical and noncanonical AIMs is that they are typically located in IDPRs. Our AF2-multimer predictions are also in agreement with this view, as most AIM/LIR sequences that we analysed are in IDPRs (Figs 1–4).

According to our assessments, AF2-multimer correctly predicts the ATG8–AIM interface with greater confidence in some models than in others. However, a high confidence structural prediction of a complete protein is not necessary to identify an ATG8 interface, as these regions can be ordered or disordered based on their interaction partners. The top-ranked AF2-multimer models correctly predicted the functional AIM/LIRs in all tested proteins except for the M2 protein from Influenza A virus. In the case of M2, the rank 1 model had a lower confidence score than the other models, which accurately showed the LIR–LC3 binding interface (Fig 4D). Therefore, models with the highest confidence scores in regions covering the AIM/LIR residues are more informative when determining AIM/LIR–ATG8-binding interfaces.

A current limitation of AF2-multimer is that it does not provide any information about the ATG8-binding affinity of the AIM/LIR motif. For instance, phosphorylation of residues flanking the LIR/AIM have been shown to improve ATG8-binding affinity [16,17], but these cannot be determined by AF2-multimer. Nevertheless, since AF2-multimer was still able to identify AIM/LIRs without considering posttranslational modifications, comparative analysis of the side chain interactions of flanking AIM/LIR residues revealed by AF2-multimer could provide further insights into AIM/LIR specificity towards different ATG8 members.

Despite its advantages, there is still room to improve AF2-multimer since not all AIM/LIR residues can be predicted. In 3 cases, i.e., the plant autophagy cargo receptor Joka2/NBR1 and the human Calreticulin and DVL2 proteins, AF2-multimer failed to predict the experimentally suggested AIM/LIRs [35,49,53]. We cannot rule out the possibility that ATG8/LC3 binding of such proteins is facilitated by posttranslational modifications [56] and/or interactions with other proteins. Currently, such conditions are not considered by AF2-multimer. Intriguingly, the AF2-models suggest that other binding interfaces exist for these proteins. This raises the question of whether AF2-multimer can be used to predict additional, novel ATG8-binding interfaces. For instance, AF2-multimer predicted a previously unidentified LC3-binding interface in the malaria virulence factor PbUIS3, which lacks a typical LIR motif (Fig 4D). According to AF2-multimer models, the previously characterised PbUIS3 residues (D173, Y174, D175, E205, and K209) that are required for LC3 interaction are not located in the LC3-binding interface, indicating that they might indirectly affect LC3 binding. Consistent with this notion, these residues are also not conserved in other *Plasmodium* species [54], which contradicts the observations that AIM/LIR motifs are located in conserved, structurally disordered regions [14]. Thus, apart from identifying validated AIM/LIRs, AF2-multimer could be an important tool for determining new ATG8-binding interfaces, providing insights into cases where ATG8/LC3 interaction remains cryptic and developing new hypotheses that can be tested experimentally.

Here, we present an AF2-multimer–based framework for identifying AIM/LIR motifs that could be used to discover novel autophagy receptors, adaptors, or modulators as well as pathogen virulence factors that target ATG8 proteins. This framework, in turn, should help address key questions in plant and human autophagy, such as: How are specific cargoes captured and

mobilised through selective autophagy? How does specific cargo selection help organisms withstand cellular and environmental stress? How are certain genetic diseases linked to defects in selective autophagy? What are the determinants of degradative versus secretory autophagy? And how do pathogens manipulate autophagy to promote diseases? The AI-guided quest for the discovery of autophagy modulators will substantially accelerate our understanding of the molecular basis of autophagy in all kingdoms of life.

## Materials and methods

### Structural analysis using AF2

We analysed a total of 51 proteins, from 11 different species, that interacted with 12 different members of the ATG8-family of various species. Data can be found in S1 Data. Sequences were obtained from UniProt. Homologs of proteins studied were found using BLAST [57] and their sequences were aligned by Clustal Omega [58]. AF2-multimer [26] was used through a subscription to the Google Colab (https://colab.research.google.com/github/sokrypton/ColabFold/blob/main/AlphaFold2.ipynb#scrollTo=svaADwocVdwl) following guidelines on the document [25]. Superposition of AlphaFold2 predictions on known structures was performed using the align command in PyMOL (The PyMOL Molecular Graphics System, Version 2.3.5 Schrödinger, LLC). Where indicated, predictions of AIM/LIR interactions are coloured according to the AlphaFold2 produced per-residue confidence metric called the local distance difference test (pLDDT), which corresponds to the model's predicted score on the lDDT-C$\alpha$ metric [59]. The scale ranges from 0 to 100, where 100 corresponds to values of highest confidence. Models were coloured by this score in PyMOL using a script generated by J. Murray and D. Pretorius (Imperial College London).

### Plasmid constructs

RFP- and GFP-ATG8CL used in this study were published previously [49]. C-terminal GFP-tagged ATI2 construct was generated by Gibson assembly of PCR fragment amplified from *S. tuberosum* cDNA into EcoRV linearized pKGC3S GFP expression vector. The DNA fragment for ATI2 AIM1 peptide was custom synthesised (GeneWiz) and inserted into the pKGN3S vector to generate an N-terminal GFP expression construct using Gibson assembly. ATI2-AIM1_(17)AEVA(20), ATI2-AIM2_(265)AQVA(268), and ATIAIM1+2 mutant fragments were generated using site-directed mutagenesis (SDM) polymerase chain reaction (PCR) amplification from wild-type construct. Templates were then eliminated by 1-h Dpn-I (New England Biolabs) restriction digestion at 37°C, and the PCR products of mutants were inserted into EcoRV-digested C-terminal GFP expression vectors using Gibson assembly. ATG3 AIM peptide DNA fragments were custom synthesised (GENEWIZ) and inserted into the pKRN3S vector to generate an N-terminal RFP expression construct using Gibson assembly. Expression vectors were transformed via electroporation into Agrobacterium strain GV3101 for plant expression.

### Plant material and growth conditions

*N. benthamiana* plants were grown and maintained in a greenhouse with high light intensity (16 h light/8 h dark photoperiod) at 22 to 24°C.

### Co-immunoprecipitation experiments and immunoblot analysis

Proteins were transiently expressed by agroinfiltration *N. benthamiana* leaves and harvested 2 days post agroinfiltration. Protein extraction, purification, and western blot analysis steps were

performed as described previously [60]. In brief, 2 g of leaf tissue was grinded in 4 mL extraction buffer (25 mM Tris-HCl (pH 7.5), 1 mM EDTA, 150 mM NaCl, 10% glycerol (v/v), and 10 mM DTT, 0.1% IGEPAL in the presence of plant protease inhibitor cocktail (Sigma-Aldrich) and 2% polyvinylpolypyrrolidone). Following centrifugation at 17,000 g for 2 times 20 min (filtration in between), the resultant supernatant was incubated with RFP beads (Chromotek) for 1 h at 4˚C. Beads were washed at 800 x g 3 times prior to elution with an elution buffer (4xLaemmli Buffer (BioRad) and DTT). Proteins were eluted at 70˚C for 5 min. After gel electrophoresis and transfer to PVDF membrane, polyclonal anti-GFP (Chromotek) produced in rabbit, monoclonal anti-RFP (Chromotek) produced in mouse, and monoclonal anti-GFP (Chromotek) produced in rat were used as primary antibodies. For secondary antibodies, anti-mouse antibody (Sigma-Aldrich), anti-rabbit (Sigma-Aldrich), and anti-rat (Sigma-Aldrich) antibodies were used.

## Confocal microscopy and quantitative analysis of GFP-ATG8CL autophagosome puncta

Imaging was performed using Leica Stellaris 5 inverted confocal microscope (Leica Microsystems) using 63× water immersion objective. All microscopy analyses were carried out on live leaf tissue 3 days after agroinfiltration. Leaf discs of *N. benthamiana* were cut and mounted onto Carolina observation gel (Carolina Biological Supply Company) to minimise the damage. Specific excitation wavelengths and filters for emission spectra were set as described previously [61]. GFP and RFP probes were excited using 488 and 561 nm laser diodes and their fluorescent emissions detected at 495 to 550 and 570 to 620 nm, respectively. Maximum intensity projections of Z-stack images were presented in each figure. Image analysis was performed using Fiji and Inkscape. For quantifying GFP-ATG8 puncta, maximum-intensity projection of each confocal microscopy image was generated using Fiji [62]. The total cell surface area was measured after removing, if there was any, the stomata or pavement cells from the images using the "Freehand selections" tool of Fiji, if the cells were not expressing GFP-ATG8. Since the signal from the ATG8 puncta has been higher than the cytoplasmic-ATG8, "Find maxima" tool was used to measure the number of puncta, with the "prominence" setting between 45 and 60 depending on the signal intensity of each image. The relative number of puncta per 10,000 $\mu m^2$ was then calculated to plot a graph. The statistical significance of any differences in means was found using the Wilcoxon test in R, because the Shapiro–Wilk test showed the data did not follow normal distribution. Three asterisks (***) in the box plot indicates that the *p* value is smaller than 0.001.

## Supporting information

**S1 Fig. AF2-multimer calculated confidence for prediction of experimentally validated canonical AIM/LIRs. (A–F)** AF2 and experimentally verified canonical AIM/LIR ATG8-family interactions, shown in Fig 1C–1H, coloured based on AF2-calculated prediction score; pLDDT. Blue indicates regions of a protein with a score of over 90, meaning a very high confidence prediction. Scores between 70 and 90 are represented with light blue, which accounts for a high confidence score. Scores between 50 and 70 are considered low (orange) and anything below 50 is a very low confidence prediction (yellow). **(G)** Calreticulin does not interact with LC3A via its canonical LIR, which seems to be positioned away from LC3A (surface representation–grey). However, Calreticulin may interact with the C-terminal region of LC3A. **(H)** Joka2 AIM residue ILE824 is localised in the alpha-helix and no ATG8 interaction is observed. **(I)** DVL2 may interact with LC3A LIR pockets via a different canonical AIM, **(J)** as the experimentally studied LIR was observed away from LC3A and structured into a beta-

sheet. Calreticulin, Joka2, and DVL2 are coloured based on the AF2-calculated prediction confidence score, pLDDT.
(TIFF)

**S2 Fig. AF2-multimer can distinguish between functional AIM/LIRs of NEDD4 and ATI2.** **(A)** The pockets of LC3B remain empty when both LIRs of NEDD4 are truncated. **(B)** Upon the reversal of LIR -1 and -2 positions, AF2-multimer favours LIR1 (previously LIR2). **(C, D)** AF2-multimer prediction confidence of the NEDD4 AIM1 and AIM2 in complex with LC3B. **(E)** AF2-multimer prediction confidence for ATI2 AIM1 in complex with ATG8CL. **(F)** ATI2 AIM2 cannot occupy ATG8CL pockets when AIM1 is truncated. **(G)** Predicted GABARAP and RTN3 interaction by LIR3; FEVI. NEDD4 and ATI2 in panels (B–F) are coloured based on the AF2-calculated prediction confidence score; pLDDT. Blue indicates regions of a protein with a score of over 90, meaning a very high confidence prediction. Scores between 70 and 90 are represented with light blue, which accounts for a high confidence score. Scores between 50 and 70 are considered low (orange) and anything below 50 is a very low confidence prediction (yellow).
(TIFF)

**S3 Fig. AF2-multimer determined confidence scores for C53 and cross-kingdom interactions with ATG8-family members.** **(A, B)** Wild-type C53 preferentially interacts with ATG8A pockets via the canonical AIM (cAIM) formed by (304)YxxV(307) and this region of the protein within the complex is predicted at a high confidence. **(C)** Once the canonical AIM (cAIM—green) of C53 (pink) is mutated, a complex with ATG8A is formed by a shuffled AIM (sAIM); **(F)** however, this is not at high confidence, although it matches experimental observations. **(E)** RavZ, **(F)** IAVM2 model ranked third, with highest confidence at LIR region, **(G)** rank one model for IAVM2 and **(H)** PexRD54 complexes with respective ATG8-family members. **(I)** The suggested LIR of UIS3 (DYD) resides in an alpha-helix. All panels are coloured based on the AF2-calculated prediction confidence score; pLDDT. Blue indicates regions of a protein with a score of over 90, meaning a very high confidence prediction. Scores between 70 and 90 are represented with light blue, which accounts for a high confidence score. Scores between 50 and 70 are considered low (orange) and anything below 50 is a very low confidence prediction (yellow) (see Methods for further details).
(TIFF)

**S1 Data. ATG8-interacting proteins and their sequences used in this study.**
(XLSX)

**S2 Data. All numerical data points supporting Fig 3G.**
(CSV)

**S3 Data. R-studio code used for the statistical analysis of numerical data points in Fig 3G.**
(RTF)

**S1 Raw images. Uncropped images for blots used in Figs 2G, 2H and 3F.**
(PDF)

## Acknowledgments

Daniella Pretorius (Imperial College London) from Dr. James Murray's group (Department of Life Sciences, Imperial College London, UK) wrote the script to generate Alphafold confidence score-based colouring in PyMOL. We thank Prof. Sophien Kamoun (The Sainsbury Laboratory, Norwich, UK) for encouraging us to move forward with our initial findings.

## Author Contributions

**Conceptualization:** Yasin Tumtas, Doryen Bubeck, Tolga O. Bozkurt.

**Data curation:** Tarhan Ibrahim, Virendrasinh Khandare, Federico Gabriel Mirkin, Doryen Bubeck, Tolga O. Bozkurt.

**Formal analysis:** Tarhan Ibrahim, Virendrasinh Khandare, Federico Gabriel Mirkin, Yasin Tumtas, Doryen Bubeck, Tolga O. Bozkurt.

**Funding acquisition:** Doryen Bubeck, Tolga O. Bozkurt.

**Investigation:** Tarhan Ibrahim, Virendrasinh Khandare, Federico Gabriel Mirkin, Yasin Tumtas, Doryen Bubeck, Tolga O. Bozkurt.

**Methodology:** Tarhan Ibrahim, Virendrasinh Khandare, Federico Gabriel Mirkin, Yasin Tumtas, Doryen Bubeck, Tolga O. Bozkurt.

**Project administration:** Doryen Bubeck, Tolga O. Bozkurt.

**Resources:** Federico Gabriel Mirkin, Yasin Tumtas, Doryen Bubeck, Tolga O. Bozkurt.

**Software:** Tarhan Ibrahim, Yasin Tumtas, Doryen Bubeck, Tolga O. Bozkurt.

**Supervision:** Doryen Bubeck, Tolga O. Bozkurt.

**Validation:** Tarhan Ibrahim, Virendrasinh Khandare, Federico Gabriel Mirkin, Yasin Tumtas, Tolga O. Bozkurt.

**Visualization:** Tarhan Ibrahim, Virendrasinh Khandare, Federico Gabriel Mirkin, Yasin Tumtas, Tolga O. Bozkurt.

**Writing – original draft:** Tarhan Ibrahim, Doryen Bubeck, Tolga O. Bozkurt.

**Writing – review & editing:** Tarhan Ibrahim, Virendrasinh Khandare, Federico Gabriel Mirkin, Yasin Tumtas, Tolga O. Bozkurt.

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
