## [Editor Report · Decision Letter 0]

4 Oct 2022

Dear Dr Bozkurt, 

Thank you for submitting your manuscript entitled "AlphaFold2-multimer guided high accuracy prediction of typical and atypical ATG8 binding motifs" for consideration as a Discovery Report by PLOS Biology.

Your manuscript has now been evaluated by the PLOS Biology editorial staff, as well as by an academic editor with relevant expertise, and I am writing to let you know that we would like to send your submission out for external peer review.

IMPORTANT: Whilst submitted as a Discovery Report, we feel that your article would be better suited as a 'Methods and Resources' Article. We ask that you please tick this article type when filling out the metadata during re-submission (see below).

Before we can send your manuscript to reviewers, we need you to complete your submission by providing the metadata that is required for full assessment. To this end, please login to Editorial Manager where you will find the paper in the 'Submissions Needing Revisions' folder on your homepage. Please click 'Revise Submission' from the Action Links and complete all additional questions in the submission questionnaire.

Once your full submission is complete, your paper will undergo a series of checks in preparation for peer review. After your manuscript has passed the checks it will be sent out for review. To provide the metadata for your submission, please Login to Editorial Manager (https://www.editorialmanager.com/pbiology) within two working days, i.e. by Oct 06 2022 11:59PM.

Kind regards,

Richard

Richard Hodge, PhD

Associate Editor, PLOS Biology

rhodge@plos.org

PLOS

---

## [Decision Letter · Decision Letter 1]

4 Nov 2022

Dear Dr Bozkurt,

Thank you for your patience while your manuscript "AlphaFold2-multimer guided high accuracy prediction of typical and atypical ATG8 binding motifs" went through peer-review at PLOS Biology. Your manuscript has now been evaluated by the PLOS Biology editors, an Academic Editor with relevant expertise, and by two independent reviewers.

The reviews are pasted below. As you will see, the reviewers are positive about the method and we are pleased to offer you the opportunity to address the comments from the reviewers in a revision that we anticipate should not take you very long. We will then assess your revised manuscript and your response to he reviewers' comments with our Academic Editor aiming to avoid further rounds of peer-review, although might need to consult with the reviewers, depending on the nature of the revisions.

In addition, I would be grateful if you could address the following editorial and data-related requests (A-C):

(A) You may be aware of the PLOS Data Policy, which requires that all data be made available without restriction: http://journals.plos.org/plosbiology/s/data-availability. For more information, please also see this editorial: http://dx.doi.org/10.1371/journal.pbio.1001797

- Supplementary files (e.g., excel). Please ensure that all data files are uploaded as 'Supporting Information' and are invariably referred to (in the manuscript, figure legends, and the Description field when uploading your files) using the following format verbatim: S1 Data, S2 Data, etc. Multiple panels of a single or even several figures can be included as multiple sheets in one excel file that is saved using exactly the following convention: S1_Data.xlsx (using an underscore).

- Deposition in a publicly available repository. Please also provide the accession code or a reviewer link so that we may view your data before publication.

Figure 3G

(B) Please also ensure that each of the relevant figure legends in your manuscript include information on *WHERE THE UNDERLYING DATA CAN BE FOUND*, and ensure your supplemental data file/s has a legend.

(C) We require the original, uncropped and minimally adjusted images supporting all blot and gel results reported in the following figures:

Figure 2G-H, 3F

We will require these files before a manuscript can be accepted so please prepare and upload them now. Please carefully read our guidelines for how to prepare an and upload this data: https://journals.plos.org/plosbiology/s/figures#loc-blot-and-gel-reporting-requirements.

We expect to receive your revised manuscript within 2 months. Please email us (plosbiology@plos.org) if you have any questions or concerns, or would like to request an extension. 

**IMPORTANT - SUBMITTING YOUR REVISION**

*Resubmission Checklist*

*Published Peer Review*

*PLOS Data Policy*

Kind regards,

Richard

Richard Hodge, PhD

Associate Editor, PLOS Biology

rhodge@plos.org

REVIEWS:

Reviewer #1: In this manuscript, Ibrahim et al. examined the utility of the protein structure prediction tool AlphaFold-Multimer (AF2-multimer) to identify ATG8-interacting motifs (AIMs) respective LC3-interacting regions (LIRs). Using a set of nearly three dozen well-characterized ATG8/LC3-interacting proteins the authors showed that AF2-multimer is able to detect the vast majority of established canonical and non-canonical AIM/LIRs. By combining in silico motif-swapping and -deletion approaches with biochemical experiments on the plant protein ATI2 the authors confirmed that AF2-multimer correctly maps canonical AIMs/LIRs among several potential candidates. The authors went on to showed that this approach also allows the identification of a previously uncharacterized, non-canonical-type AIM in plant ATG3. Lastly, the authors expanded their analysis to ATG8/LC3-binding virulence factors from different human and plant pathogens and found that AF2-multimer correctly predicted their experimentally validated AIMs/LIRs. Overall, the work of Ibrahim and colleagues elegantly shows that AF2-multimer cannot only be employed to identify validated AIMs/LIRs but might also be a useful tool to map new ATG8/LC3 binding sites. However, a few critical issues remain.

1) While the authors show AF2-multimer can identify functional AIM/LIR in proteins with up to two AIMs/LIRs candidates (e.g., NEDD4 and ATI2), a number of proteins have far more LIRs than that. The authors should examine AF2-multimer's performance on proteins with truly multiple AIMs/LIRs such as RTN4 (Grumati et al. Elife 2017 (PMID: 28617241)).

2) To test the limits/capabilities of AF2-muzltimers, the authors should examine the non-canonical LIR in ATG12 (Kaufmann et al. Cell 2013 PMID: 24485455).

3) All immunoblots are missing molecular weight markers. Please add them. 

Reviewer #2: Ibrahim et al have used the AlphaFold2-multimer software tool to perform protein modelling to predict LIR/AIM motifs in proteins bidnign to ATG8 family proteins.

I was amazed how succesful the authors have been using this tool to predict LIR motifs of both known and unknown LIR-containing proteins and also of non-canonical LIR motifs.

This is a really large step forward in predicting ATG8-interacting motifs that the autophagy research community will benefit from!

It is also of general interest as a strategy and tool to find motifs binding to specific proteins. 

I am very positive to this nice work and do not have major criticisms.

The Introduction does not mention ATG8ylation and could perhaps refer to a review on this (PMID: 34671813, PMID: 34527862).

It would perhaps also be pertinent to refer to an extensive review on ATG8s and LIR motifs (PMID: 31310766) in the second pragraph of the INTRODUCTION.

It was not clear to me the exact sequences used to do the modeling and prediction in each case. This needs to be shown in supplemental material.

I did not see a supplemental Table 1 referred to in the paper, only an excel file with core LIR sequences.

It is important to know if a 300 amino acid sequence can be placed in and give an accurate prediction of one LIR sequence for example or if shorter sequences must be probed sequentially

to get the best results.

---

## [Decision Letter · Decision Letter 2]

15 Dec 2022

Dear Dr Bozkurt,

Thank you for the submission of your revised Methods and Resources Article "AlphaFold2-multimer guided high accuracy prediction of typical and atypical ATG8 binding motifs" for publication in PLOS Biology. Please accept my apologies for the delays you have experienced during the re-review process. 

On behalf of my colleagues and the Academic Editor, Anne Simonsen, I am pleased to say that we can accept your manuscript for publication, provided you address any remaining formatting and reporting issues. These will be detailed in an email you should receive within 2-3 business days from our colleagues in the journal operations team; no action is required from you until then. Please note that we will not be able to formally accept your manuscript and schedule it for publication until you have completed any requested changes.

PRESS

Kind regards, 

Richard

Richard Hodge, PhD

Associate Editor, PLOS Biology

rhodge@plos.org

PLOS
